# Counterfactual Groups to Assess Vaccine or Treatment Efficacy in HIV Prevention Trials in High-Risk Populations in Uganda

**DOI:** 10.3390/vaccines13080844

**Published:** 2025-08-08

**Authors:** Andrew Abaasa, Yunia Mayanja, Zacchaeus Anywaine, Sylvia Kusemererwa, Eugene Ruzagira, Pontiano Kaleebu

**Affiliations:** 1MRC/UVRI & LSHTM Uganda Research Unit, Entebbe P.O. Box 49, Uganda; yunia.mayanja@mrcuganda.org (Y.M.); zacchaeus.anywaine@mrcuganda.org (Z.A.); sylvia.kusemererwa@mrcuganda.org (S.K.); eugene.ruzagira@mrcuganda.org (E.R.); pontiano.kaleebu@mrcuganda.org (P.K.); 2London School of Hygiene & Tropical Medicine (LSHTM), London WC1E 7HT, UK

**Keywords:** counterfactual, vaccine, treatment, efficacy, HIV-prevention, trial, active, controls

## Abstract

Background: Assessment of efficacy in HIV prevention trials remains a challenge in the era of widespread use of active controls. We investigated use of counterfactual groups to assess treatment efficacy. Methods: We used data from placebo arms of two previous HIV prevention efficacy trials (Pro2000 vaginal microbicide trial, 2005–2009: ISRCTN64716212 and dapivirine vaginal ring trial, 2013–2016: NCT01539226) and four observational cohorts (two in each of the periods; (a) during the conduct of a simulated HIV vaccine efficacy trial (SiVET), 2012–2017, and (b) prior to SiVET (2005–2011)) and compared HIV prevention efficacy trial targeted outcomes with SiVETs. SiVET participants were administered a licensed hepatitis B vaccine at 0, 1 and 6 months mimicking an HIV vaccine efficacy trial schedule. Participants were tested for HIV quarterly for one year. The probability of the SiVET assignment conditioned on the measured participants’ baseline characteristics were estimated using propensity scores (PS) and matched between SiVET and placebo arm of trials. Similar calculations were repeated for observational cohorts in the pre- and during SiVET periods. We compared HIV incidence rate ratio (IRR) between SiVET and the trials or observational data before and after PS matching. Results: This analysis involved data from 3387 participants; observational cohorts before SiVET 1495 (44.2%), placebo arms of previous trials 367 (10.8%), observational cohorts during SiVET conduct 953 (28.1%) and SiVETs 572 (16.9%). Before propensity score matching (PSM), there were significant imbalances in participants’ baseline characteristics between SiVET, and all the other studies and HIV incidence was lower in SiVET. After PSM, the participants’ characteristics were comparable. The HIV incidence in SiVET was similar to that in the previous trial, IRR = 1.01 95% CI: 0.16–4.70), *p* = 0.968, and observational data during SiVET, IRR = 0.74, 95% CI 0.34–1.54), *p* = 0.195, but much lower compared to the observational data pre-SiVET, IRR = 0.48, 95% CI: 0.20–1.04), *p* = 0.023. Conclusions: PSM can be used to create counterfactual groups from other data sources. The best counterfactual group for assessing treatment effect is provided by data collected in the placebo arm of previous trials followed by that from observational data collected concurrently to the current trial (SiVET). Even with PSM, observational data collected prior to the current trial may overestimate treatment effect.

## 1. Background

Globally, HIV infections continue to occur mostly in Sub-Saharan Africa (SSA). In 2020 alone, 1.5 million people became newly infected with HIV, 58% in SSA [1]. The new infections in SSA are largely driven by the suboptimal adherence to available HIV prevention interventions [2,3]. An effective and affordable HIV vaccine could be the only hope to minimize non-adherence and end HIV acquisition. HIV candidate vaccines have to go through assessments in efficacy trials. The availability of many HIV prevention interventions such as pre-exposure prophylaxis (PrEP), post-exposure prophylaxis (PEP), medical male circumcision (MMC) and condoms among others make the design of efficacy trials challenging because of the ethical requirements of active controls [4,5]. Provision of these interventions in a trial diminishes the background HIV incidence observed in the population from which subjects enrolled in efficacy trials are obtained, hence requiring long-term follow-up or extremely large sample sizes. Efficacy trial investigators have proposed use of HIV incidence obtained from observational controls or earlier trials [3,6,7] or Averted Infection Ratio (AIR) (i.e., dividing the rate difference between hypothetical placebo and experimental arms by the rate difference between hypothetical placebo and active control arms) to estimate comparator HIV incidence [8]. Each approach has its challenges; for instance, participants that join clinical trials are different from those that do not, making a comparison to observational controls biased [9]. Secondly, earlier clinical trials may have been conducted in a different population or may suffer from the effect of time and improvements in healthcare standards [3]. Those proposing AIR suggest use of HIV incidence from a run-in period cohort or that from another trial or epidemiological surveillance, all which might introduce bias because of the differences in time and/or population.

In our previous analysis [3], we showed how propensity scores (PS) and propensity score matching (PSM) statistical approaches that attempt to estimate treatment effect [10,11] can be used to create counterfactual-trial-arms-enabling estimation of unbiased hypothetical arm HIV incidence. This paper [3] considered data collected in simulated HIV vaccine efficacy trials (SiVETs) [9] drawing controls from observational cohorts conducted in the same population prior to or during the SiVETs [9]. The HIV incidence data are available from multiple sources such as observational data in other populations and at different time points or even from placebo arms of previous trials in diverse populations.

To expand on our previous work, we considered all the available data from four observational cohorts, i.e., two prior to the SiVETs [12,13], and two during the SiVETs [9] and placebo arms of two randomized controlled trials (RCTs) prior to and during SiVETs conducted in diverse populations and geographical locations [14,15]. In this paper, we created three counterfactual groups as follows; (a) observational cohorts (OBC) prior to SiVETs, (b) OBC during SiVETs period and (c) placebo arms of data available from randomized controlled trials. Lastly, we estimated and compared HIV incidence in SiVETs to that in the observational cohorts and RCTs before and after propensity score matching. This is aimed at proposing counterfactual groups as opposed to a placebo arm that can be used to assess treatment/vaccine effect in the future HIV prevention trials with active controls.

## 2. Methods

Design: The data used in this analysis were collected in four observational cohorts (OBC); (a) prior to SiVET: (i) HIV serodiscordant couples 2005–2009, (ii) first fisherfolk (FF) 2009–2011; (b) SiVET concurrent period: (iii) female sex workers (FSW), 2010–2017, (iv) second fisherfolk 2012–2014, and two placebo arms of randomized controlled trials (RCTs) of HIV prevention products; (c) prior to SiVET: (v) pro2000 vaginal gel trial, 2005–2009; (d) SiVET concurrent period: (vi) dapivirine ring trial 2013–2016; (e) two simulated HIV vaccine efficacy trials (SiVETs) 2012–2017 in the FF and FSW populations. Further details are shown in Figure 1.

### 2.1. Description of the OBCs, Trials and SiVETs

(i)HIV serodiscordant couples OBC: This OBC was established at the MRC/UVRI and LSHTM Uganda Research Unit field in Masaka district and recruited participants in a known HIV serodiscordant couple relationship [13]. The primary aims of this cohort were the following: (i) creation of a recruitment source population for future HIV vaccine efficacy trials and (ii) determine HIV incidence (three-monthly).(ii)First fisherfolk OBC: This OBC was established at clinics located within five fishing communities along the shoreline of lake Victoria; two in Entebbe subdistrict about 30 km south of Kampala, Uganda’s capital, and three in Masaka district about 100 km southwest of Kampala [12,16]. The co-primary aims of this first fisherfolk OBC were as follows: (i) to determine HIV incidence (six-monthly) and (ii) assess annual retention.(iii)Female sex workers OBC: This cohort recruited FSWs from hotspots of sex work business including bars, night clubs, restaurants, etc., in Kampala city [9]. The primary aims of this cohort were similar to those of the fisherfolk population cohorts and HIV testing was performed every three months.(iv)Second fisherfolk OBC: Unlike the first (ii) above where participants were seen at clinics established in each of the five participating fishing communities, this required that participants travel to a clinic established at the MRC/UVRI and LSHTM Unit Station in Masaka town about 50 km from fishing communities [9,17]. This cohort had similar aims as the first fisherfolk cohort.(v)PRO2000 vaginal microbicide gel trial (Microbicides Development Programme (MDP301) trial), trial registration ISRCTN64716212: This randomized, double blind, placebo-controlled phase III clinical trial recruited HIV negative women in a known HIV serodiscordant heterosexual couple relationship in a 1:1:1 ratio to 2% PRO2000, 0.5% PRO2000, or matching placebo gel groups [14,18]. The primary aim was assessing efficacy and safety of the different PRO2000 gel formulations against HIV transmission in women. None of the gel formulations showed effectiveness against HIV acquisition. In this study we used only data from the placebo gel group.(vi)Dapivirine vaginal ring trial, registration NCT01539226: This randomized, double blind, placebo-controlled phase III trial recruited HIV negative women at risk of HIV infection in a 2:1 ratio to receive vaginal rings containing either 25 mg of dapivirine or placebo [15,19]. The primary aim was to assess whether the dapivirine vaginal ring was safe and effective in preventing HIV in women compared to placebo ring. The trial was conducted at MRC/UVRI and LSHTM Unit field clinic in Masaka district. We used only data from the placebo vaginal ring group.(vii)Simulated HIV vaccine efficacy trials (SiVETs): These two sequential and similar SiVETs (first in the fisherfolk in Masaka district (2012–2014) and secondly in the female sex workers (2014–2017) in Kampala) recruited participants at high risk of HIV infection from the second fisherfolk and female sex workers cohorts, respectively. Participants received a commercially licensed hepatitis B vaccine (ENGERIX-BTM GlaxoSmithKline Biologicals Rixensart, Rixensart, Belgium) following the standard schedule of 0, 1 and 6 months and under conditions that mimicked an HIV vaccine efficacy trial with extra follow-up visits for up to 12 months. The primary aim was to assess retention in a trial environment and train trial staff. In these studies, HIV testing was performed every three months. Details of both SiVETs have been previously published [9,20,21].

Data stratification: The different studies were stratified into two mutually exclusive periods, (1) pre-SiVET and (2) SiVET concurrent periods. Pre-SiVET included the data collected in (a) OBCs of the first fisherfolk and HIV serodiscordant couples cohorts, and (b) the placebo arm of the PRO2000 vaginal gel trial. SiVET concurrent period included the data collected in (c) OBCs of the second fisherfolk and female sex workers cohorts and (d) the placebo arm of the dapivirine ring trial. More details and layout of these cohorts and trials are provided in Figure 1.

### 2.2. Key Evaluations in This Analysis

(i)We compared participants’ baseline characteristics between SiVETs and (a) all the observational cohorts pre-SiVET, (b) observational cohorts in the SiVET concurrent period and (c) repeated a and b above for trials, all before and after propensity score matching (PSM).(ii)We further made the comparisons in (i) above for HIV incidence.

### 2.3. HIV Testing

A single HIV antibody rapid test was performed using Alere Determine HIV-1/2 (Alere Medical Co Ltd., Matsuhidai, Matsudo-shi, Chiba, Japan). All rapid HIV-positive results were confirmed by two parallel-enzyme-linked immunosorbent assay (ELISA) tests (Murex Biotech Limited, Dartford, UK, and Vironostika, BioMérieux, Boxtel, The Netherlands). Western Blot (Cambridge Biotech, Worcester, MA, USA) confirmed any discordant results. All HIV testing was performed at MRC/UVRI and LSHTM Uganda Research Unit clinical diagnostic laboratories except for HIV seroconverters and those with discordant HIV results in the two trials that underwent a confirmation PCR assay for HIV RNA on stored samples at the same HIV testing reference laboratory in South Africa since these trials were part of larger multicounty trials.

### 2.4. Statistical Methods

All the analyses were performed in STATA version 15.0 (Stata Corp, College Station, TX, USA). Data were extracted from the different databases, formatted to common data structure and appended to one dataset. We divided the studies into two groups, (a) observational cohorts and (b) clinical trials, and into two periods, (i) pre-SiVET and (ii) SiVET concurrent periods. Because the primary aim of this paper was to compare SiVET participants to those in each of the other groups, we transformed participants’ baseline characteristics that were common to all studies to the same variable values before appending the different datasets. With categorizations are shown in the tables, the variables considered in this analysis were as follows: sex, age, education level, religion, marital status, alcohol use, number of sexual partners, having a new sexual partner, condom use, self-reported genital discharge and genital ulcer disease.

Propensity score (PS) and propensity score matching (PSM): We fitted logit models in which assignment into SiVET was regressed on baseline covari-ates of a comparator study (OBCs or RCTs), first in the pre-SiVET and secondly in the SiVET concurrent periods. This was aimed at estimating the probability of SiVET assignment conditioned on documented participants’ baseline covariates. In each period, we considered the variables indicated in Figure 2 to perform PSM. We performed a 1:1 propensity score matching without replacement to within a caliper of 0.2 between SiVET and the respective comparator (OBC or RCT) in the pre-SiVET and SiVET concurrent periods. This was aimed at providing balanced (comparable) participant baseline characteristics between SiVET and the respective comparator groups. A balance in characteristics was achieved with a standardized mean difference of <0.20. Propensity score matching that assumes a caliper width of 20% of the pooled standard deviation of the logit of the propensity score is thought to provide superior performance in the estimation of treatment effects [22]. Furthermore, <20% difference in covariates after propensity score matching is considered an indication of good matching [23]. Participants in SiVETs that we could not find matches for in the comparator groups were excluded from the matched analysis. Consequently, the dapivirine ring trial had only 67 participants and this could not allow meaningful PSM between SiVET and RCT data in the concurrent period. We compared participants’ baseline characteristics between SiVETs and OBCs and RCTs in both periods before and after PSM using Chi-square tests. We illustrated graphically the standardized differences in participants’ covariates between SiVETs and OBCs or RCTs before and after PSM in both periods. To compare the primary outcome of this analysis, i.e., HIV incidence between SiVETs and comparator groups before and after PSM, we estimated HIV incidence as number of participants newly diagnosed with HIV divided by the total person–years at risk (PYAR) expressed as per 100 PYAR. PYAR were estimated as sum of the time from the enrolment date into the respective study to the last HIV seronegative test result date or an estimated date of HIV infection for participants that seroconverted. The date of HIV infection was estimated as a random, multiple-imputation date between the last HIV seronegative test result date and the first HIV seropositive test result date. Rate ratios and 95% confidence intervals were obtained using a Poisson regression model with robust standard errors. We further conducted a sensitivity analysis leveraging on a larger dataset, i.e., combining the data from observational cohorts and clinical trials compared with SiVETs.

Figure 2 shows standardized percent bias in the participants’ baseline covariates between SiVETs and OBCs or placebo arm of RCT in both pre-SiVET and SiVET concurrent periods. Pre-SiVET; between (a) OBCs and SiVETs and (b) placebo arm of RCT and SiVETs. SiVET concurrent period; between (c) OBCs and SiVETs. From these graphs, it can be deduced that before PSM in all periods, the standardized percent bias across covariates between SiVETs and OBCs or RCT varied over a wide range (−100% to + 150%) and between −10% and + 10% after PSM.

## 3. Results

The analysis involved data from 3387 participants and is distributed as follows: (a) prior to SiVETs: 1495 (44.1%) from observational cohorts and 300 (8.9%) from Pro2000 HIV prevention trial; (b) SiVETs concurrent: 953 (28.1%) from observational cohorts, 67 (2.0%) from the dapivirine ring HIV prevention trial and 572 (16.9%) from SiVETs, Figure 1.

### 3.1. Baseline Characteristics

Observational cohorts prior to SiVET, before and after PSM: Before PSM (i.e., covariate balance), comparing SiVET to OBC, almost all baseline characteristics differed significantly between SiVETs and OBCs, *p* < 0.05. For instance, SiVETs had more females (64.2% vs. 40.5%), participants with ≥2 sexual partners (66.1% vs. 29.5%), alcohol users (62.8% vs. 51.4%) and fewer participants with genital discharge (22.7% vs. 61.8%) and genital sores (18.4% vs. 64.6%), in the table below. Before PSM, the standardized differences in participants’ characteristics between OBCs and SiVETs ranged from 3.9% in the religion covariate to 100% in the genital sores covariates. After PSM, participants’ baseline characteristics were comparable between OBCs and SiVETs and the standardized differences reduced to between 0.0% in the education level and 12.1% in the age group covariates, Table 1.

Placebo arm of RCT prior to SiVET, before and after PSM: Before PSM, comparing participants’ baseline characteristics between SiVETs and the placebo arm of RCT, all participants’ baseline characteristics differed between SiVETs and the RCT, *p* < 0.05, Table 2. For instance, SiVETs enrolled more of participants with secondary or more education (39.8% vs. 16.3%), Muslim faith (24.8% vs. 15.7%), ≥2 sexual partners (73.3% vs. 13.3%) and those using condom during sex (68.1% vs. 13.8%) in the table below. Before PSM, the standardized differences in participants’ baseline characteristics between SiVETs and placebo arm of the RCT ranged from 19.2% in alcohol use covariate to 152.2% in the number of sexual partners’ covariate. After PSM, the baseline characteristics were comparable between SiVETs and the RCT and the standardized differences reduced to between 0.0% in the genital sores and alcohol use covariates and 7.4% in the numbers of sexual partners’ covariate, Table 2.

OBC in the SiVET concurrent period, before and after PSM: Before PSM, comparing SiVET to OBC in the concurrent period, SiVET had more participants that were male (35.8% vs. 14.4%), aged >34 years (35.1% vs. 21.4%), had ≥secondary education (30.9% vs. 18.3%) and were currently/previously married (73.4% vs. 65.8%) in the table below. Furthermore, SiVETs had more participants reporting low-HIV-risk behaviors including non-alcohol use (37.2% vs. 28.9%), none or one sexual partner (33.9% vs. 20.8%) and having no new sexual partners (14.5% vs. 5.1%) in the table. Before PSM, the standardized difference in the participants’ baseline characteristics between OBCs and SiVETs ranged from 0.8% in the religion covariate to 51.1% in the sex covariate. After PSM, all baseline covariates were comparable between the two studies, (*p* > 0.05), and the standardized differences reduced to between 0.5% in the genital discharge and alcohol use covariates and 10.8% in the condom use covariate, Table 3.

Results of the sensitivity analysis: After PSM, more matches were obtained and covariate balance was achieved for all variables except genital discharge (standardized mean difference 0.22). On the contrary, after PSM, the difference in the HIV incidence between observational-plus-trial data and SiVETs became wider, 6.8 95% CI: 4.9–9.4 vs. 3.9 95% CI: 2.3–6.6. This could be because of the differences in (a) time of conducting studies and/or (b) combining data from different designs (observational cohorts and trials).

### 3.2. HIV Incidence

Generally, HIV incidence was lower in SiVETs than OBCs and placebo arm of RCT before PSM but only achieved statistical significance (*p* = 0.033) when comparing incidence in SiVETs to that in OBCs in the SiVET concurrent period, as shown in the table below. After PSM, the trend of lower HIV incidence in SiVETs relative to that in OBCs narrowed and only achieved statistical significance (*p* = 0.023) when comparing incidence in SiVETs to that in OBCs in the pre-SiVET period, Table 4.

## 4. Discussion

In this analysis, we investigated the application of PS and PSM to (a) placebo arm data from previous RCTs or (b) OBC data collected prior to or in the SiVETs concurrent period to create counterfactual groups (non-randomized but comparable to SiVETs) and compared HIV incidence between these groups and SiVETs. We found that before PSM, in all studies and all periods, participants’ baseline characteristics differed from those in SiVETs. HIV incidence in the placebo arm of RCT and OBCs in all periods was higher than in SiVETs but only achieved statistical significance in the OBC in the SiVET concurrent period. Surprisingly, even when the OBCs in the concurrent period were the recruitment source for SiVETs, participants’ characteristics differed significantly between the two groups. Literature shows that participants that volunteer to join clinical trials are often different from those that do not and the HIV incidence is likely to differ [9,24,25]. Participants with high-risk behavior are less likely to volunteer as trial participants [3]. Compared to SiVETs, OBCs in both periods were dominated by participants with characteristics that have been previously associated with high risk of HIV infection such as young age, low education level, having a new sexual partner, low condom use with new sexual partners, genital discharge and genital ulcer disease infection [12,13,26,27,28,29]. With many HIV prevention products becoming available, research ethics will require comparison to active controls [4,5]. This will make design of trials challenging or require long-term follow-up of participants and/or a large sample size to obtain a sufficient number of the targeted outcomes. To our knowledge, no study has compared HIV incidence in the placebo arm of previous RCTs to that in HIV vaccine efficacy trials. Even then, such a comparison would still be affected by differences in participant characteristics and improvements in healthcare including HIV prevention and care service. Our comparison of participants in SiVETs to those in the placebo arm of HIV prevention RCT conducted prior to SiVETs showed significant differences in participants’ baseline characteristics before PSM. The HIV incidence was higher in the placebo arm of HIV prevention RCTs though the difference did not achieve statistical significance.

PS and PSM, statistical techniques for creating counterfactual study groups, have been previously used to assess unbiased treatment effects [10,11]. These techniques were employed in this analysis to create three counterfactual groups (i) from placebo arm of RCTs, (ii) OBCs prior to SiVET and (iii) OBCs in the SiVET concurrent period, and to compare HIV incidence between each of these groups and SiVET. After PSM, participants’ baseline characteristics in all studies and periods were comparable to those in the SiVETs. Comparing OBCs to SiVET, results suggest that SiVET participation reduced HIV incidence by 26% and 52% in the SiVETs concurrent period and pre-SiVET period, respectively. No differences in HIV incidence were observed between the SiVETs and the placebo arm of HIV prevention RCT conducted prior to the SiVETs. The literature indicates that the trial environment provides HIV risk reduction care beyond that in the observational studies [24,25]. This could reduce HIV incidence even in absence of an efficacious investigational product. In our SiVETs we provided a hepatitis B vaccine that is unlikely to have a positive effect on the risk of HIV infection. However, SiVET participants received healthcare beyond that in the observational cohort in both periods and HIV incidence was lower. For instance, we provided counseling on multiple or new sexual relationships, condom use with a new/causal sexual partner, offered free condoms with timely refills whenever required and active diagnosis and treatment of sexually transmitted and other genital infections. While most of these were common to both OBCs and SiVETs, OBC participants received condoms on request and no active diagnosis and treatment of STIs and other genital infections were performed. Furthermore, SiVET participants visited the trial clinic more regularly, hence accessing these interventions more.

Our analysis benefits from several strengths; availability of data from the placebo arms of HIV prevention RCTs and OBCs prior to SiVETs and in the SiVETs concurrent periods with adequate follow-up time, large sample sizes to provide matches and data from several populations in diverse geographical locations and concurrent conduct of OBCs and SiVETs. The results provide strong evidence that PSM can be used to create counterfactual groups to assess treatment effect with better comparison provided by data from placebo arms of previous randomized controlled trials.

Our analysis limitations included the following; in SiVETs, we provided hepatitis B vaccine after informing participants that this vaccine prevents hepatitis B infection and not HIV. This could have encouraged more participation by participants considering themselves at high risk of hepatitis B infection creating selection differences between SiVETs and observational cohort. Studies considered in this analysis were conducted at varying times, meaning there might have been some unmeasured positive effects of improvements in healthcare on HIV incidence and variation in the quality of the data collected. Additionally, we considered only variables that were common to all studies, therefore we could have missed some important confounders. Even with these possible differences, results suggest that SiVETs HIV incidence is lower than that in observational cohorts irrespective of when the data were collected. Furthermore, data from the placebo arms of previous HIV prevention RCTs used in this analysis was exclusively from female participants. Hence, the results of the comparison to previous HIV prevention trials are mostly generalizable to efficacy trials of female volunteers.

## 5. Conclusions

Characteristics of volunteers enrolled into efficacy trials are different from those in the source population observational cohorts or placebo arms of previously conducted efficacy trials. This difference leads to lower HIV incidence in efficacy trials than that in the source observational cohorts or previous trials even in absence of an efficacious investigational product. Therefore, data from observational cohorts conducted prior to or in the concurrent period as efficacy trials or that from placebo arms of previous trials’ may not be used as a control arm to assess treatment effectiveness without adjustment. Our results suggest that propensity score and propensity score matching can be used to remove imbalance in the participants’ characteristics in all comparator groups, creating counterfactual groups that can be used to assess treatment effect. However, in our analysis, after creating counterfactual groups, HIV incidence in SiVETs was only similar to that in the placebo arms of previous trials but 26% and 52% lower than that in the observational cohorts in the concurrent and pre-SiVET periods, respectively. Taking the entire results of this analysis, counterfactual arms derived from the placebo arms data of previous trials provide the best control arms to assess treatment effect. Where such data are not available, counterfactual arms derived from observational data in the efficacy trial concurrent period would be preferred but HIV incidence would have to be adjusted downwards by 25% to approximate that in the actual trial control arm. In absence of data from both of these sources, counterfactual arms derived from observational data prior to the efficacy trials can be used to assess treatment effect. However, HIV incidence from such counterfactual data would have to be adjusted downwards by approximately 50%.

## Figures and Tables

**Figure 1 vaccines-13-00844-f001:**
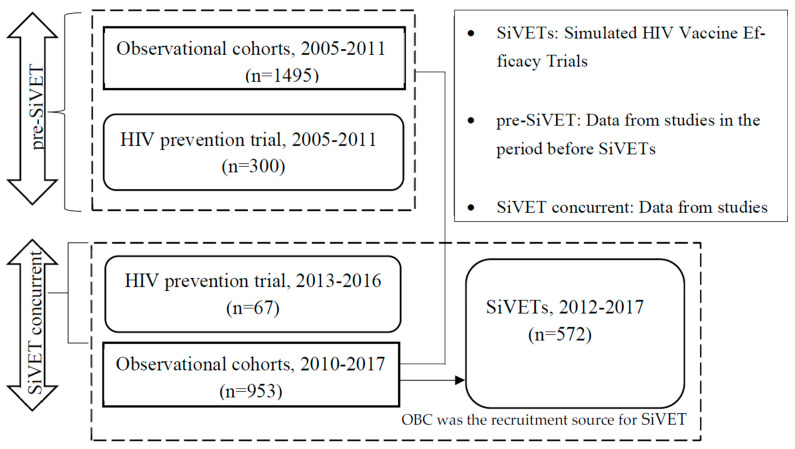
Observational cohorts and placebo arm of HIV prevention trials pre-SiVET and during SiVET periods.

**Figure 2 vaccines-13-00844-f002:**
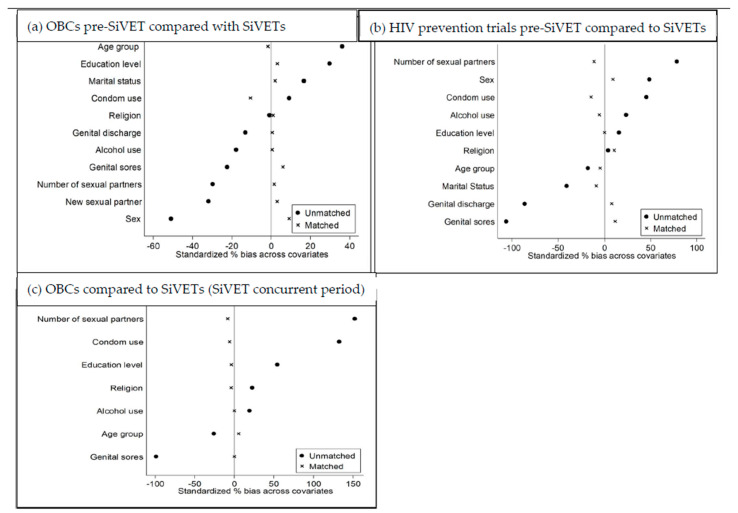
Standardized differences across covariates: before and after covariate matching.

**Table 1 vaccines-13-00844-t001:** Comparison of participants’ baseline characteristics between SiVETs and OBCs in the pre-SiVET period both before and after covariates balance.

	Before PSM				After PSM			
Variables	OBCs *n* = 1495 (%)	SiVETs *n* = 572 (%)	*p*-Value	Std (Diff)	OBCs *n* = 319 (%)	SiVETs *n* = 319 (%)	*p*-Value	Std (Diff)
Sex			<0.001	0.487			0.263	0.089
Male	889 (59.5)	205 (35.8)			188 (58.9)	174 (54.5)		
Female	606 (40.5)	367 (64.2)			131 (41.1)	145 (45.5)		
Age (years)			<0.001	0.240			0.311	0.121
18–24	400 (26.8)	173 (30.2)			82 (25.7)	80 (25.1)		
25–30	399 (26.7)	198 (34.6)			86 (27.0)	103 (32.3)		
31+	696 (46.5)	201 (35.2)			151 (47.3)	136 (42.6)		
Education			0.001	0.156			1.000	0.000
Primary/none	1136 (76.0)	395 (69.1)			243 (76.2)	243 (76.2)		
Secondary +	359 (24.0)	177 (30.9)			76 (23.8)	76 (23.8)		
Religion			0.426	0.039			0.183	0.106
Christian	1164 (77.9)	436 (76.2)			255 (79.9)	241 (75.5)		
Muslim	331 (22.1)	136 (23.8)			64 (20.1)	78 (24.5)		
Number of sexual partners			<0.001	0.787			0.178	0.107
0–1	1054 (70.5)	194 (33.9)			158 (49.5)	175 (54.9)		
2+	441 (29.5)	378 (66.1)			161 (50.5)	144 (45.1)		
Alcohol use			<0.001	0.232			0.474	0.057
No	727 (48.6)	213 (37.2)			141 (44.2)	150 (47.0)		
Yes	768 (51.4)	359 (62.8)			178 (55.8)	169 (53.0)		
Genital discharge			<0.001	0.861			0.327	0.078
No	571 (38.2)	442 (77.3)			238 (74.6)	227 (71.2)		
Yes	924 (61.8)	130 (22.7)			81 (25.4)	92 (28.8)		
Genital sores			<0.001	1.063			0.158	0.112
No	529 (35.4)	467 (81.6)			238 (74.6)	222 (69.6)		
Yes	966 (64.6)	105 (18.4)			81 (25.4)	97 (30.4)		

OBC—Observational cohort, SiVET—Simulated vaccine efficacy trial, PSM—Propensity score matching, Std (diff)—Standardized difference.

**Table 2 vaccines-13-00844-t002:** Comparison of participants’ baseline characteristics between SiVETs and the placebo arm of RCT in the pre-SiVET period both before and after covariates balance.

	Before PSM				After PSM			
Variables	RCT *n* = 369 (%)	SiVET *n* = 367 (%)	*p*-Value	Std (Diff)	RCT *n* = 119 (%)	SiVET *n* = 119 (%)	*p*-Value	Std (Diff)
Age (years)			<0.001	0.297			0.907	0.057
18–24	83 (22.5)	105 (28.6)			33 (27.7)	30 (25.2)		
25–30	108 (29.3)	138 (37.6)			36 (30.3)	37 (31.1)		
31+	178 (48.2)	124 (33.8)			50 (42.0)	52 (43.7)		
Education			<0.001	0.543			0.774	0.037
Primary/none	309 (83.7)	221 (60.2)			84 (70.6)	86 (72.3)		
Secondary +	60 (16.3)	146 (39.8)			35 (29.4)	33 (27.7)		
Religion			0.002	0.227			0.747	0.042
Christian	311 (84.3)	276 (75.2)			94 (79.0)	96 (80.7)		
Muslim	58 (15.7)	91 (24.8)			25 (21.0)	23 (19.3)		
Number of sexual partners			<0.001	1.522			0.566	0.074
0–1	320 (86.7)	98 (26.7)			83 (69.7)	87 (73.1)		
2+	49 (13.3)	269 (73.3)			36 (30.3)	32 (26.9)		
Alcohol use			0.009	0.193			1.000	0.000
No	157 (42.5)	122 (33.2)			54 (45.4)	54 (45.4)		
Yes	212 (57.5)	245 (66.8)			65 (54.6)	65 (54.6)		
Condom use			<0.001	1.324			0.676	0.054
No	318 (86.2)	117 (31.9)			80 (67.2)	83 (69.7)		
Yes	51 (13.8)	250 (68.1)			39 (32.8)	36 (30.3)		
Genital sores			<0.001	0.992			1.000	0.000
No	141 (38.2)	300 (81.7)			68 (57.1)	68 (57.1)		
Yes	228 (61.8)	67 (18.3)			51 (42.9)	51 (42.9)		

RCT—Randomized controlled trial, SiVET—Simulated vaccine efficacy trial, PSM—Propensity score matching, Std (diff)—Standardized difference.

**Table 3 vaccines-13-00844-t003:** Comparison of participants’ baseline characteristics between SiVETs and OBCs in the SiVET concurrent period both before and after covariates balance.

	Before PSM				After Propensity PSM			
Variables	OBCs *n* = 953 (%)	SiVETs *n* = 572 (%)	*p*-Value	Std (Diff)	OBCs *n* = 442 (%)	SiVETs *n* = 442 (%)	*p*-Value	Std (Diff)
Sex			<0.001	0.511			0.200	0.086
Male	137 (14.4)	205 (35.8)			130 (29.4)	113 (25.6)		
Female	816 (85.6)	367 (64.2)			312 (70.6)	329 (74.4)		
Age (years)			<0.001	0.362			0.832	0.041
18–24	431 (45.2)	173 (30.3)			145 (32.8)	152 (34.4)		
25–30	318 (33.4)	198 (34.6)			164 (37.1)	156 (35.3)		
31+	204 (21.4)	201 (35.1)			133 (30.1)	134 (30.3)		
Education			<0.001	0.298			0.651	0.030
Primary/none	779 (81.7)	395 (69.1)			324 (73.3)	318 (71.9)		
Secondary +	174 (18.3)	177 (30.9)			118 (26.7)	124 (928.1)		
Religion			0.874	0.008			0.875	0.011
Christian	723 (75.9)	436 (76.2)			337 (76.2)	335 (75.8)		
Muslim	230 (24.1)	136 (23.8)			105 (23.8)	107 (24.2)		
Marital status			0.002	0.167			0.765	0.020
Single never married	326 (34.2)	152 (26.6)			127 (28.7)	123 (27.8)		
Married (current or previous)	627 (65.8)	420 (73.4)			315 (71.3)	319 (72.2)		
Alcohol use			0.001	0.179			0.943	0.005
No	275 (28.9)	213 (37.2)			144 (32.6)	143 (32.4)		
Yes	678 (71.1)	359 (62.8)			298 (67.4)	299 (67.6)		
Number of sexual partners			<0.001	0.298			0.818	0.015
0–1	198 (20.8)	194 (33.9)			117 (26.5)	114 (25.8)		
2+	755 (79.2)	378 (66.1)			325 (73.5)	328 (74.2)		
New sexual partner			<0.001	0.319			0.627	0.033
No	49 (5.1)	83 (14.5)			39 (8.8)	35 (7.9)		
Yes	904 (94.9)	489 (85.5)			403 (91.2)	407 (92.1)		
Condom use			0.084	0.092			0.108	0.108
No	428 (44.9)	231 (40.4)			149 (33.7)	172 (38.9)		
Yes	525 (55.1)	341 (59.6)			293 (66.3)	270 (61.1)		
Genital discharge			0.014	0.131			0.939	0.005
No	682 (71.6)	442 (77.3)			327 (74.0)	326 (73.8)		
Yes	271 (28.4)	130 (22.7)			115 (26.0)	116 (26.2)		
Genital sores			<0.001	0.223			0.365	0.061
No	689 (72.3)	467 (81.6)			354 (80.1)	343 (77.6)		
Yes	264 (27.7)	105 (18.4)			88 (19.9)	99 (22.4)		

OBC—Observational cohort, SiVET—Simulated vaccine efficacy trial, PSM—Propensity score matching, Std (diff)—Standardized difference.

**Table 4 vaccines-13-00844-t004:** HIV incidence before and after covariate balance in the pre-SiVET and SiVET concurrent periods.

			SiVETs		OBCs		RCT	Risk	
Period	Method	HIV +	Incidence (95%CI)	HIV +	Incidence (95%CI)	HIV +	Incidence (95% CI)	Incidence Rate Ratio (95% CI)	*p*-Value
**OBCs**-pre-SiVET	Before PSM	17	3.5 (2.2–5.6)	93	4.7(3.8–5.7)		-	0.75 (0.42–1.27)	0.136
	After PSM	9	3.2 (1.6–6.1)	28	6.6 (4.6–9.6)		-	0.48 (0.20–1.04)	0.023
**RCT**-pre-SiVET	Before PSM	10	3.4 (1.8–6.4)	-	-	23	4.2 (2.8–6.3)	0.82 (0.35–1.79)	0.305
	After PSM	3	2.94 (0.9–9.1)	-	-	6	2.93 (1.3–6.5)	1.01 (0.16–4.70)	0.968
**OBCs**-SiVET concurrent	Before PSM	17	3.5 (2.2–5.6)	39	5.9 (4.3–8.1)		-	0.59 (0.31–1.07)	0.033
	After PSM	14	3.9 (2.3–6.6)	20	5.3 (3.4–8.2)		-	0.74 (0.34–1.54)	0.195

OBC—Observational cohort, RCT—Randomized controlled trial, SiVET—Simulated vaccine efficacy trial, PSM—Propensity score matching, CI—confidence interval, PYAR—person–years at risk.

## Data Availability

The MRC/UVRI and LSHTM Uganda Research Unit encourages data sharing and has a published (https://apps.mrcuganda.org/mrcdatavisibility/Home/) data sharing process. This policy summarizes the conditions under which data collected by the Unit can be made available to other bona fide researchers, the way in which such researchers can apply to have access to the data and how data will be made available if an application for data sharing is approved. Access is granted through a publicly available data campus (https://datacompass.lshtm.ac.uk/view/creators/), accessed on 30 July 2025. Should any of the other researchers need to have access to the data from which this manuscript was generated, the processes to access the data are well laid out in the policy. The corresponding and other co-author emails have been provided and could be contacted anytime for any clarifications and/or support to access the data.

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
