# Peer review of "Counterfactual Groups to Assess Vaccine or Treatment Efficacy in HIV Prevention Trials in High-Risk Populations in Uganda"

_vaccines, 2025, doi:10.3390/vaccines13080844_

Round 1

Reviewer 1 Report

Comments and Suggestions for Authors

Andrew Abaasa and colleagues have presented a paper testing whether simulated HIV vaccine trials (SIVETs) could be compared to other study types using matched baseline characteristics to estimate HIV incidence when placebo arms are not feasible. The authors used data from 3387 participants from two HIV prevention trials, four observational cohorts, and two SIVETs. The studies were broken down into 2 time periods concurrent to or prior to SIVET trials. SIVET participants received a licensed hepatitis B vaccine at 0,1, and 6 months similar to HIV vaccine schedules. Participants were tested for HIV using a standard algorithm and confirmed by PCR. A propensity score matching (PSM) method was applied to control for differences in demographics and sexual behavior across groups. The authors found that participants in SIVETs had significantly different baseline risk profiles and lower HIV incidence than in observational cohorts and clinical trials. After PSM, incidence in SIVETs was similar to the placebo arm of a previous clinical trial. The authors found the incidence was lower in SIVET compared to concurrent observational cohorts, and significantly lower than pre-SIVET cohorts. The best comparator group was from prior trial placebo arms.

The authors raise and attempt to address an important area in HIV prevention. As more modalities become available to prevent HIV infection, the statistical and ethical implication in the design of future efficacy studies becomes challenging. This paper suggests and alternative method to having an active placebo arm in these studies and instead suggest estimating treatment effects using existing data. While the idea is important there are several issues that should be addressed.

  1. The execution of this study dates back from 2005 through 2017. Since the execution of these studies, even more progress has been made in prevention efforts and implementation. The authors should discuss how current and future modalities may impact using this strategy. The authors could also discuss some of the recent literature on clinical trial design in this context, such as work from Holly Janes (PMID 36938334)
  2. The authors were unable to use the contemporaneous clinical trial to compare to SIVET. What value does it provide to the paper? Further, the authors break up distinct cohorts and studies, but more data usually give better results. Can you combine all observational and clinical trial data and compare the combination of OBC and clinical trial data to SIVET? The time factor may limit the utility of such an approach.
  3. The strongest comparator group was the placebo arm of the PRO2000 vaginal microbicide gel trial which was exclusively female. This limitation needs more discussion in the paper. How do the authors reconcile this with the mixed sex composition of the SIVET? Can the authors perform an analysis of just the women from SIVET compared to PRO2000?
  4. Conclusion section? Should lines 367-369, “Conclusions This section is not mandatory but can be added to the manuscript if the discussion is unusually long or complex”, be deleted? Or did you mean to put in a conclusion.
  5. Figure 1 is quite confusing. What is the arrow from OBC to SIVET showing? Should the bracket for the SIVET concurrent extend to OBC? Why is there a line connecting the two OBC? Does the box with the bullets provide any additional insight into the figure that is not provided from the text of the paper?
  6. Figure 2 is effective in showing the differences between pre- and post-PSM but could benefit from clearer labels. Could you color code this to emphasize the differences between the groups?
  7. One minor point is that “deffered” on lines 226 and 282 should be “differed”

Reviewer 2 Report

Comments and Suggestions for Authors

The authors have addressed the very important issue of finding appropriate control groups for the assessment of potential HIV vaccines. The issue is compounded by the exposure of potential controls to HIV prevention interventions, which may render analysis of the impact of potential vaccines on HIV incidence difficult. In this manuscript, the authors have built upon previously published studies to demonstrate that the generation of propensity scores (PS) and subsequent propensity score matching (PSM) can be used to create counterfactual trial groups that could enable an unbiased base for the estimation of HIV incidence in future HIV vaccine trials. The groups included for analysis in the paper include two placebo groups from studies to assess antimicrobial therapies on HIV prevention, and two observational cohorts from simulated HIV vaccine efficacy trials (SiVETs) prior to and during the vaccination period. As this manuscript is heavily based upon statistical analysis, it is essential that the appropriate statistical approach is used. The authors have presented a detailed explanation of their approach, which is a strength of the manuscript. While the data presented in each of the tables is very dense, it provides a strong basis for justifying the approach and the use of this approach in future studies. Of particular importance is the data presented in the tables comparing the participants’ baseline characteristics after PSM analysis. Overall, this is an important study that will provide a strong guideline for future studies on the efficacy of HIV vaccines in the face of pre-existing preventative therapies so that appropriate control comparison groups can be generated.

Reviewer 3 Report

Comments and Suggestions for Authors

Dear Authors,

This manuscript investigates the use of counterfactual groups derived via propensity score matching (PSM) from previous HIV prevention trials and observational cohorts to estimate HIV incidence in simulated HIV vaccine efficacy trials (SiVETs). The topic is timely and conceptually important, especially in light of ethical and logistical limitations of placebo-controlled HIV prevention trials. However, several issues limit the clarity, rigor, and reproducibility of the work, and substantial revisions are necessary to improve the manuscript.

  1. Clarity of Study Aims and Hypotheses
    The central hypothesis and objectives are not clearly stated in the abstract or the introduction. It remains unclear whether the primary aim is to validate PSM for counterfactual construction or to propose a practical alternative to placebo arms in future HIV prevention trials. Please revise the abstract and introduction to clearly articulate the research objectives and underlying hypotheses.
  2. The manuscript contains considerable repetition, particularly in the descriptions of cohorts and trial structures. Background information is reiterated across the Introduction and Methods sections. This content should be streamlined to enhance readability.
  1. Statistical Rigor of the Methodological Details
    While PSM is described as the primary method, the rationale for covariate selection, matching procedures, and balance diagnostics is not well explained. Please, Specify the variables included in the propensity score models and justify their selection. Provide balance diagnostics (e.g., standardized mean differences, graphical assessments) and clarify whether matching was done with or without replacement.
    - Please clearly state which statistical models or tests were used to calculate incidence rate ratios (IRRs). Additionally, indicate whether robust variance estimators or clustered standard errors were applied, especially given the integration of data from multiple sources.
    - The manuscript does not include a sensitivity analysis or address the issue of unmeasured confounders. This is a significant limitation given the reliance on observational data. Please conduct a sensitivity analysis to test the robustness of PSM estimates, or at minimum, discuss this limitation explicitly. A supplementary methodological table detailing matching procedures and diagnostics would also enhance transparency.
  1. The conclusion that placebo arms from earlier RCTs are superior to observational cohorts as counterfactual controls is not fully supported by the data. Furthermore, the recommended downward adjustments of 25% or 50% in HIV incidence estimates from observational data are presented without methodological justification. Please clarify whether these adjustments are empirically derived or hypothetical, and revise the discussion accordingly.
  2.  The manuscript requires thorough language editing. Numerous grammatical and typographical errors are present (e.g., “counterfacual” instead of “counterfactual”, “deminishes” instead of “diminishes”). A comprehensive editorial review is recommended.
  3. Figure 1 lacks a clear structure and visual hierarchy. Relationships between cohorts, SiVETs, and trial timelines are not intuitively represented. Arrows or connectors should be added to illustrate temporal or population flow. A revised figure with better formatting and labeling would significantly enhance the reader's understanding of the study design and data structure.

This manuscript addresses an important methodological issue, but revisions are necessary to improve clarity, methodological transparency, and interpretability. I encourage the authors to consider the above suggestions to strengthen the quality and impact of the work.

Comments on the Quality of English Language

Numerous grammatical and typographical errors are present (e.g., “counterfacual” instead of “counterfactual”, “deminishes” instead of “diminishes”). A comprehensive editorial review is recommended.

Round 2

Reviewer 1 Report

Comments and Suggestions for Authors

Comments were addressed somewhat satisfactorily.  The authors could considering writing a statement about the additional results of the combined observational data and clinical trials compared to SiVETs and the confounding reason this approach may not be suitable. Figure 2 formatting is off with the labels of the panels obscuring the data and content of each panel.
